# An Optimal Sensor Layout Using the Frequency Response Function Data within a Wide Range of Frequencies

**DOI:** 10.3390/s22103778

**Published:** 2022-05-16

**Authors:** Eun-Taik Lee, Hee-Chang Eun

**Affiliations:** 1Department of Architectural Engineering, Chung-Ang University, Seoul 06974, Korea; etlee@cau.ac.kr; 2Department of Architectural Engineering, Kangwon National University, Chuncheon 24341, Korea

**Keywords:** optimal sensor placement, model reduction, modal assurance criterion, effective independence, proper orthogonal decomposition, frequency response function

## Abstract

This study presents iterative optimal sensor placement (OSP) techniques using the modal assurance criterion (MAC) and the effective independence (EI) algorithm. The algorithms use the proper orthogonal mode (POM) extracted from the frequency response functions (FRFs) of dynamic systems within a wide range of frequencies. The FRF-based OSP method proposed in this study has the merit of reflecting dynamic characteristics, unlike the mode shape-based method. Evaluating the MAC values and the EI indices at each iteration, the DOFs of low contribution to the objective function of candidate sensor DOFs are deleted from master DOFs and moved to slave DOFs. This process is repeated until the sensor number corresponds with the master DOFs. The validity of the proposed methods is illustrated in an example, the sensor layouts by the proposed methods are compared, and the layout inconsistency between the MAC and the EI techniques is analyzed.

## 1. Introduction

A structure can be diagnosed by tracking the response data collected from the sensors attached to it. Although it is possible to obtain response information by attaching a large number of sensors to the structure, certain maintenance and economic aspects pose difficulties. It is necessary to perform measurements with the optimal number and location of sensors. Optimal sensor placement (OSP) involves designing a sensor layout for diagnosing the structural performance with a prescribed number of sensors. The OSP technique is important for structural health monitoring, damage detection, system identification, performance evaluation, etc.

Many OSP techniques have been proposed, and most take the mode shape-based iteration process for mode shape reconstruction. The effective independence (EI) method proposed by Kammer [1,2] for sensor design is one of the most widely used OSP techniques. This method optimizes the sensor locations depending on the contribution to the linear independence between mode shapes containing significant factors. The low contribution sensor locations of the linear independence are removed one by one from the initial candidate sensor locations. The iteration is repeated until the number of prescribed sensors is reached. The final sensor locations become the positions used to maximize the determinant of the Fisher information matrix and to minimize its condition number. Jiang et al. [3] proposed an alternative formula for the EI algorithm, and they mentioned the physical significance of the re-orthogonalization of the mode shape through QR decomposition. Friswell and Castro-Triguero [4] investigated the relationship between the EI approach and the linear independence of the mode shapes. Lee and Eun [5] compared the OSP algorithms of the reduced-order model using the constraint condition between the master and the slave modes from the target modes and various objective functions.

The modal assurance criterion (MAC) matrix is expressed by the correlation between two modal vectors. The off-diagonal elements of the MAC matrix are considered as indices to evaluate an objective function. The off-diagonal elements of the MAC matrix require a scalar product between two mode vectors. Thus, if the angle between the two vectors is large, the off-diagonal element is small, indicating lower correlation between the two vectors. The minMAC-based OSP method utilizes this concept. Carne and Dohrmann [6] exhibited that the MAC-based sensor design procedure of starting small and adding sensors is more efficient than starting with the entire DOFs and reducing the number of sensors. He et al. [7] presented an OSP technique to combine minMAC in order to improve the modal energy, and an improved adaptive genetic algorithm in order to enhance the computation efficiency. Fu and Yu [8] proposed a sensor optimization method to decide the sensor numbers by MAC and the sensor location using a single parenthood genetic algorithm. Brehm et al. [9] enhanced the conventional MAC algorithm with additional physical information regarding modal strain energy. Jung et al. [10] provided a genetic-algorithm-based sensor optimization method using the sensor positions and the MAC. Morlier et al. [11] provided an OSP method by maximizing the trace of Auto MAC in order to reconstruct a truncated modal basis of mode shapes.

One of the measurement data types is the frequency response function (FRF) dataset. The FRF can be estimated by an impact hammer test, and the energy is propagated from the impact point. The FRF data measured from experimental work are utilized to estimate the characteristics of a dynamic system. Measured FRF data can be utilized to predict the parameter values of a dynamic system. This indicates the importance of optimal sensor design for collecting more accurate information on the system.

The existing mode shape-based OSP methods are inefficient because they use data limited to the low-frequency region. The FRF-based OSP technique has merit in establishing the sensor layout by the data within the high-frequency range, unlike the mode shape-based OSP technique. Rama Mohan Rao et al. [12] presented an FRF-based OSP technique using principal components extracted from FRF data, and they compared it with the EI technique based on mode shapes. Garcia-Palencia et al. [13] evaluated an FRF-based model updating algorithm using experimentally collected data, and they presented a protocol for measurement selection and a regularization technique. Alqam and Dhingra [14] presented a dynamic load identification method through structural FRFs using accelerometer and strain gage sensors. Yuan and Zhang [15] presented an OSP approach to design much fewer sensors by using the condition number of the modal shape matrix, including the FRF.

The proper orthogonal decomposition (POD) method has been widely used in engineering fields. It is utilized to decompose structural responses into a set of spatial functions, in a similar manner to principal component analysis and singular value decomposition. The sensors are placed in locations where energy is relatively concentrated. Kerschen et al. [16] studied the POD to extract the order reduction and the feature extraction modes. Nimityongskul and Kammer [17] developed an OSP technique based on frequency response data in the mid-frequency range using principal component analysis. They mentioned that the optimal sensors are at positions that represent the independence and the signal strength of the dominant principal directions. Cherng [18] proposed methods to predict modal parameter identification, such as natural frequencies, mode shapes, and damping ratios based on the singular value decomposition method. The goal of this paper is to present OSP techniques based on MAC and EI approaches using POMs extracted from measured FRFs. The approaches are iterative methods that reduce the number of candidate sensor locations and the system order, as shown in Figure 1. The dynamic system order is reduced using modal coordinates in order to apply the generalized inverse relationship between master and slave mode shapes. Evaluating the MAC values and the EI indices at each iteration, the low-contributing objective function master sensor DOFs are deleted and moved to the slave DOFs. The iteration proceeds until the prescribed sensor number coincides with the master DOFs. The validity of the proposed method is illustrated in designing the optimal sensor layout on a truss structure. It is shown that the FRF-based OSP methods proposed in this study reflect dynamic characteristics within a wide range of frequencies. The example compares the final sensor layouts between the MAC and the EI techniques, and it analyzes their inconsistency.

## 2. Optimal Sensor Design

### 2.1. POD and Model Reduction

The dynamic equation of motion for a linear dynamic system approximately discretized for *n* DOFs in the time domain can be written as:
(1)
$$M\ddot{u}+C\dot{u}+Ku=F\left(t\right)$$

where **M**, **K**, and **C** denote the 
n×n
 analytical mass, stiffness, and damping matrices derived through the finite element model, respectively; 
$u={\left[\begin{array}{c} u_{1}\;\;\;u_{2}\;\;\;\cdots \;\;\;u_{n} \end{array}\right]}^{T}$
; 
$F\left(t\right)$
 is the 
n×1
 external excitation vector.

The mode shape matrix 
ϕ
 is obtained by the characteristic equation of the undamped free vibration of the system in the frequency domain. We assume the 
n×r

$\left(n>r\right)$
 incomplete modal matrix with smaller modes than the system order and *s*

$\left(s\geq r\right)$
 target sensors as an optimal sensor number. Then, we evaluate the objective function, and lesser contributing DOFs of the candidate sensor are deleted and moved from the master to the slave DOFs. Greater contributing DOFs are retained in the master. Initially, the DOF with a small value of the first mode shape is sorted to the slave DOF, and the others become the candidate sensor locations.

Expressing the mode shape matrix as the master and slave mode shape matrix, the modal transformation of the dynamic responses can be written as:
(2)
$$u\left(t\right)=\left[\begin{array}{c} u_{m}\left(t\right)\\ u_{s}\left(t\right) \end{array}\right]=\left[\begin{array}{c} \phi_{m}\\ \phi_{s} \end{array}\right]q\left(t\right)$$

where 
$\phi_{m}$
 and 
$\phi_{s}$
 denote the 
m×r
 master and 
$\left(n-m\right)\times r$
 slave mode shape matrix, respectively; 
$u_{m}\left(t\right)$
 and 
$u_{s}\left(t\right)$
 are the 
m×1
 master and 
$\left(n-m\right)\times 1$
 slave displacement vector, respectively; and 
$q\left(t\right)$
 is the 
r×1
 modal displacement vector. The number of master DOFs *m* is *n* − 1 at the first iteration for tracing the OSP.

Solving the first part of Equation (2) with respect to the modal displacement vector 
$q\left(t\right)$
 with the help of a generalized inverse solution yields:
(3)
$$q\left(t\right)=\phi_{m}^{+}u_{m}\left(t\right)$$

where the superscript ‘+’ denotes the generalized inverse, and the substitution of Equation (3) into the second part of Equation (2) leads to:
(4)
$$u_{s}\left(t\right)=\phi_{s}\phi_{m}^{+}u_{m}\left(t\right)$$


It is observed from Equation (4) that the displacement vector corresponding to the slave DOFs is expressed by the displacements of the master DOFs. The dynamic equation can be condensed by *m* dynamic equations corresponding to the master DOFs, and contains the influence on the slave modes.

Dividing the dynamic part of Equation (1) into the equations corresponding to the master and slave DOFs, we obtain:
(5)
$$\left[\begin{array}{cc} M_{mm} & M_{ms}\\ M_{sm} & M_{ss} \end{array}\right]\left[\begin{array}{c} {\ddot{u}}_{m}\\ {\ddot{u}}_{s} \end{array}\right]+\left[\begin{array}{cc} C_{mm} & C_{ms}\\ C_{sm} & C_{ss} \end{array}\right]\left[\begin{array}{c} {\dot{u}}_{m}\\ {\dot{u}}_{s} \end{array}\right]+\left[\begin{array}{cc} K_{mm} & K_{ms}\\ K_{sm} & K_{ss} \end{array}\right]\left[\begin{array}{c} u_{m}\\ u_{s} \end{array}\right]=\left[\begin{array}{c} F_{m}\\ F_{s} \end{array}\right]$$


Substituting Equation (4) into the first part of Equation (5), the reduced dynamic equation corresponding to the master DOFs is derived as:
(6)
$${\bar{M}}_{mm}{\ddot{u}}_{m}+{\bar{C}}_{mm}{\dot{u}}_{m}+{\bar{K}}_{mm}u_{m}=F_{m}$$
where 
${\bar{M}}_{mm}=M_{mm}+M_{ms}\phi_{s}\phi_{m}^{+}$
, 
${\bar{C}}_{mm}=C_{mm}+C_{ms}\phi_{s}\phi_{m}^{+}$
**,** and 
${\bar{K}}_{mm}=K_{mm}+K_{ms}\phi_{s}\phi_{m}^{+}$
. Equation (6) represents the dynamic equation by the master DOFs only.

The dynamic equation can be transformed into the equation in the frequency domain by substituting 
$u_{m}\left(t\right)=U\left(\Omega \right)e^{j\Omega t}$
 and 
$F_{m}\left(t\right)=F\left(\Omega \right)e^{j\Omega t}$
 into Equation (6). Thus, it can be expressed as:
(7)
$$\left(-\Omega^{2}{\bar{M}}_{mm}+j\Omega {\bar{C}}_{mm}+{\bar{K}}_{mm}\right)U=F$$

where 
Ω
 is the excitation frequency and 
$j=\sqrt{-1}$
. The FRF is expressed by the relationship between the complex spectrum of the response 
$\;U\left(\Omega \right)$
 and the complex spectrum of the excitation 
$F\left(\Omega \right)$
. The FRF matrix 
$H\left(\Omega \right)$
 is written as:
(8)
$$H\left(\Omega \right)=\frac{U\left(\Omega \right)}{F\left(\Omega \right)}$$


The magnitude of the FRF denotes the ratio of the magnitude of the response to the magnitude of the excitation. 
$H_{pq}$
 in the FRF matrix indicates the relationship between a displacement response at station *p* and a disturbing force at station *q* within a finite frequency interval.

The FRFs imply dynamic characteristics. The FRFs within a frequency range are transformed by the proper orthogonal modes (POMs). The POD method is utilized to find spatially orthogonal modes regarded as a Karhunen–Loeve decomposition, singular value decomposition, and principal component analysis. The POD technique is an effective method to extract an extreme dataset from the decomposition of a large number of FRFs. The orthogonal transformation is performed using the eigenvectors of the sample covariance matrix. The POD modes are described by the maximum energy corresponding to the dominant frequencies.

The 
m×s
 measured FRF matrix represents the displacement responses at *m* DOFs due to the external excitations at 
$s\left(m\geq s\geq 1\right)$
 DOFs. The data sampled within *l* frequencies are arranged in a snapshot matrix 
$H_{total}$
:
(9)
$$H_{total}={\left[H\left(\Omega_{1}\right)\;\;\;H\left(\Omega_{2}\right)\;\;\;\cdots \;\;\;H\left(\Omega_{l}\right)\right]}_{m\times \left(l\times s\right)}$$


The POD proceeds with the determination of the eigenvectors of the 
$\left(l\times s\right)\times \left(l\times s\right)$
 autocovariance matrix 
G
, written as:
(10)
$$G=H_{total}^{T}H_{total}$$


The covariance matrix 
$\left(l\times s\right)\times \left(l\times s\right)$
 **G** is a Hermitian positive semi-definite matrix representing the covariance between each pair of elements of a given vector. It possesses a complete set of orthogonal eigenvectors with corresponding nonnegative real eigenvalues. The eigenvector corresponding to the largest eigenvalue is the direction along which the dataset has the maximum variance.

The linear eigenvalue problem of Equation (10) is considered in order to find the eigenvalue such that the linear system:
(11)
$$G\psi_{k}=\lambda_{k}\psi_{k},\;\;\;\;k=1,2,\ldots ,\left(l\times s\right)$$

has a nontrivial vector, where 
$\lambda_{k}$
 and 
$\psi_{k}$
 represent the *k*th eigenvalue and the corresponding eigenvector, respectively. The eigenvalues reflect the proper orthogonal values (POVs) to denote the energies contained in different POMs. The POVs are arranged in descending order:
(12)
$$\lambda_{1}\geq \lambda_{2}\geq \cdots \geq \lambda_{\left(l\times s\right)}\geq 0$$

where the eigenvalue 
$\lambda_{1}$
 is the largest POV to include the maximum energy, and the corresponding POM is the optimal vector to characterize the snapshots. The POV can be utilized as the index to indicate the importance of the corresponding POM.

The normalized eigenvalues represent the relative kinetic energy associated with the corresponding mode. The total kinetic energy is defined as the sum of the POVs. A set of 
$h\;\left(h\leq m\right)$
 POMs 
ψ
 corresponding to the first *h* POVs ordered from the largest POV to the smallest is written as:
(13)
$$\psi_{m}={\left[\begin{array}{cc} \psi_{m}^{1}\;\;\;\psi_{m}^{2}\;\;\; & \cdots \;\;\;\psi_{m}^{h} \end{array}\right]}_{m\times h}$$

where 
$\psi_{m}^{k}=\frac{H_{total}\psi_{k}}{||H_{total\psi_{k}}||}$
, 
k=1,2,…,h


### 2.2. MAC-Based OSP Approach

The MAC analysis in this study was used to determine the similarity between two vectors of POMs in order to replace the mode shape vectors. The value in the MAC matrix is bounded between 0 and 1 to indicate the correlated mode pairs. The element of the *i*th row and the *j*th column in the MAC matrix can be written as:
(14)
$${\mathrm{MAC}}_{ij}=\frac{{\left|\psi_{m}^{i}\psi_{m}^{j}{}^{T}\right|}^{2}}{\left(\psi_{m}^{i}\psi_{m}^{i}{}^{T}\right)\left(\psi_{m}^{j}\psi_{m}^{j}{}^{T}\right)}\;\;\;i=1,2,\ldots ,m,\;j=1,2,\ldots ,m$$


If the MAC value is 1, both vectors are fully consistent POMs and indicate high correlation; if the value is near 0, they are not consistent and a indicate low correlation. The diagonals in the MAC matrix take the value 1 because both sets of POMs are identical; the off-diagonal element values are <1 because both sets are not identical. The MAC-based OSP technique designs the sensor layout by using the minimum off-diagonal elements.

The MAC value is evaluated using the following objective function:
(15)
$$s=\sum_{i=1,j=1}^{m}{\mathrm{MAC}}_{ij}^{2}\;i\neq j$$


Starting from the *m* candidate sensors consistent with the master DOFs, the DOFs with low correlation representing the maximum off-diagonal value in the MAC matrix are moved to slave DOFs. This process is repeated until it matches the optimal number of sensors.

### 2.3. EI-Based OSP Approach

The mode shape-based EI algorithm is a widely used OSP technique and positions the sensors where the signal strength is maximized. The sensor placements exclude unsuitable DOFs from candidate positions of the sensors by the iterative EI method.

The generalized displacement vector corresponding to the master DOFs in the frequency domain can be transformed by:
(16)
$$U_{m}=\psi_{m}z$$

where 
z
 is the modal displacement vector. The estimated displacement vector is contaminated by the external noise, and it is expressed by:
(17)
$${\hat{U}}_{m}=\psi_{m}\left[z-\chi_{0}\right]$$

where 
$\chi_{0}$
 represents the 
h×1 
 Gaussian vector variance matrix.

The covariance matrix **Y** between the actual and the estimated displacements for the POMs is defined as:
(18)
$$Y=E\left[\left(U_{m}-{\hat{U}}_{m}\right){\left(U_{m}-{\hat{U}}_{m}\right)}^{T}\right]={\left[{\left(\frac{\partial \mu }{\partial U_{m}}\right)}^{T}B^{-1}\left(\frac{\partial \mu }{\partial U_{m}}\right)\right]}^{-1}=F^{-1}$$
where 
μ
 represents the average vector 
m×1
 of the change amount of the POM, 
$B=\chi_{0}^{2}$
 represents the Gaussian vector variance matrix of 
m×m
, *E* denotes the expected value, and **F** is the Fisher information matrix (FIM) of 
m×m
. Assuming a constant Gaussian variance, Equation (18) can be written as:
(19)
$$E\left[\left(\hat{U}-U\right){\left(\hat{U}-U\right)}^{T}\right]={\left[\frac{1}{\chi_{0}^{2}}\psi_{m}^{}\psi_{m}^{T}\right]}^{-1}=F^{-1}$$

where 
F
 is the FIM used to express the product of the POM matrix in Equation (13) and its transposition 
$F=\psi_{m}^{}\psi_{m}^{T}$
.

The covariance matrix **Y** becomes the minimum at the value that maximizes the FIM. Conversely, the minimization of the FIM hardly reduces the covariance value. Among the indices calculated in the master DOFs, the row of master DOFs representing the minimum value is moved to the row of slave DOFs, and the process of calculating indices of FIM is repeated and calculated. Finally, the position corresponding to the master DOFs is the sensor location. The FIM within the master DOFs is computed as:
(20)
$$\mathrm{FIM}=diag\left(\psi_{m}^{}\psi_{m}^{T}\right)$$


The maximization of FIM is induced by an eigenvalue analysis of 
$\psi_{m}^{}\psi_{m}^{T}$
:
(21)
$$\left[\psi_{m}^{}\psi_{m}^{T}-\nu I\right]\xi =0$$

where 
ν
 is the 
m
 eigenvalues, and 
ξ
 is the corresponding eigenvector. The eigenvalue contribution **Q** of the eigenvalues of the FIM in the absolute identification space is written as:
(22)
$$Q=\left[\psi_{m}^{}\xi \right]⊗\left[\psi_{m}^{}\xi \right]$$

where 
⊗
 denotes the matrix product between elements. The index, which is a diagonal element of matrix **Q**, is used to evaluate the magnitude of the FIM, and if the index is low, it is not suitable as a sensor position.

By multiplying the matrix **Q** by the inverse of the eigenvalue matrix, the EI index of the candidate sensor position is obtained:
(23)
$$E_{d}=\left[\psi \xi \right]⊗\left[\psi \xi \right]\nu^{-1}\left\{1\right\}$$

where 
$E_{d}$
 is the EI index and denotes the degree of contribution to the eigenvalue at the candidate sensor position. 
$\left\{1\right\}$
 is a 
h×1
 column vector whose element is 1. If the EI index is large, the sensor position is large; and, conversely, the influence is insignificant if the EI index is small.

## 3. Numerical Example

The MAC and the EI methods proposed in this study are utilized in a truss structure presented by Sun and Buyukozturk [19]. The numerical results and the sensor layouts were compared by tracing the candidate locations during the numerical iteration. The plane truss structure, as shown in Figure 2, consisted of 27 members and 15 nodes. The structure was simply supported at both ends. Each node had two DOFs of horizontal and vertical displacement and a total of 27 displacement DOFs, except for boundary conditions. The material properties of the truss member were selected as follows: elastic modulus 
E=200 GPa
 and mass density 
$\rho =7860\;\mathrm{kg}/m^{3}$
. The elements 1, 2, 4, 6, 8, 10, 12, 14, 16, 18, 20, 22, 24, 26, and 27 had a cross-sectional area of 
$0.01{\;m}^{2}$
, and the elements 3, 5, 7, 9, 11, 13, 15, 17, 19, 21, 23, and 25 had a cross-sectional area of 
$0.005{\;m}^{2}$
.

The number of allowable sensors for measurement was set to 6 out of 27 candidate positions. The lowest six mode shape matrix was utilized as the target mode for this example. The mode shape matrix was divided into the submatrices of master and slave DOFs, and the DOFs that contributed less to the objective function were moved from master to slave DOFs. The first estimable slave DOF was chosen at node 14y to represent the smallest value of the first mode. Here, the number is the node number, and y is the vertical mode component. The iteration was performed until the master, which consisted of 26 DOFs, was reduced to six target DOFs.

The FRF data from 0.01 Hz to 50 Hz were taken in increments of 0.02 Hz by numerical simulation. The damping matrix assumed Rayleigh damping as the sum of 0.0002 times the mass matrix and 0.0001 times the stiffness matrix. Figure 3 shows the absolute value of receptance 
$H_{3x,3x}$
, which is the horizontal displacement response to the horizontal excitation at node 3 on a logarithmic scale. Here, the number represents the node number, and x and y are the horizontal and vertical directions at the node, respectively. It can be observed from the figure that the first natural frequency was 10.33 Hz and the second natural frequency was 23.36 Hz.

A set of FRF data obtained from the numerical experiment was extracted for predicting the POMs by the POD approach. The POMs and the POVs were used to calculate MAC and the EI indices. The receptance to represent all DOF responses due to the impact in the horizontal direction at node 3 was collected with increased frequency. Since *m* POM elements corresponding to the master DOFs were normalized, the norm of all DOFs became 1. Thus, this study utilized *h* = (*m* − 4) POMs for each DOF.

The sensors should be located at positions where the mechanical behavior of the entire structure can be estimated, not the local behavior of the structure. The sensors are placed at independent DOFs to achieve effective data acquisition and to improve the accuracy of damage detection, system identification, and structural maintenance.

Figure 4 compares the DOFs removed from the master DOFs in each iteration by the MAC and the EI methods. The same DOFs were removed from the master DOFs until the initial fifth iteration, as shown in the figure. After that, it can be seen that the path of the removed DOFs was different, and it gradually converged to the independent DOFs. Since both methods evaluate the degree of the contribution to the linear independence and design of the optimal sensor, a small difference in the sensor layout between the two approaches was due to the established objective function. The optimal measurement positions obtained in this study are compared in Figure 5. When comparing the results of the two approaches, the DOF 13x was consistent. Considering the geometrical symmetry except for the boundary conditions, the DOFs 3y, 5y, and 7y in the MAC-based approach can be seen to correspond to the DOFs 13y, 11y, and 9y in the EI-based approach, respectively. The DOFs 4x and 11y in the MAC-based approach were located at the nodes adjacent to the DOFs 2x and 12y in the EI-based approach, respectively. Figure 5c shows the OSPs obtained from the example provided by Sun & Buyukozturk. This example estimated the positions of eight sensors. Compared with the results of this study, two OSP algorithms in this study exhibited the same sensor positions in four DOFs. This difference in the OSPs is interpreted to be due to the difference in the algorithm, including the location of the objective function.

The number indicates the node, and the letters x and y after the number indicate the DOF in the horizontal and the vertical directions.

## 4. Conclusions

In this study, based on the POM extracted from the measured FRF data, two methods using the MAC and the EI algorithms were proposed for determining the optimal sensor layout. Optimal sensors were designed by taking FRF data in the broad frequency domain beyond the limits of the mode shape-based algorithm in the limited frequency domain. The optimal sensor layout on the truss structure was compared in the numerical experiment using the MAC and the EI methods. As a result of comparing six optimal sensor positions of a truss structure with 27 DOFs made by the two approaches proposed in this study, one position coincided, three were in geometrically symmetrical positions, and the other two were interpreted as being in close proximity. It was determined that there was an insignificant difference in the optimal sensor design made by the two approaches due to the established objective function.

## Figures and Tables

**Figure 1 sensors-22-03778-f001:**
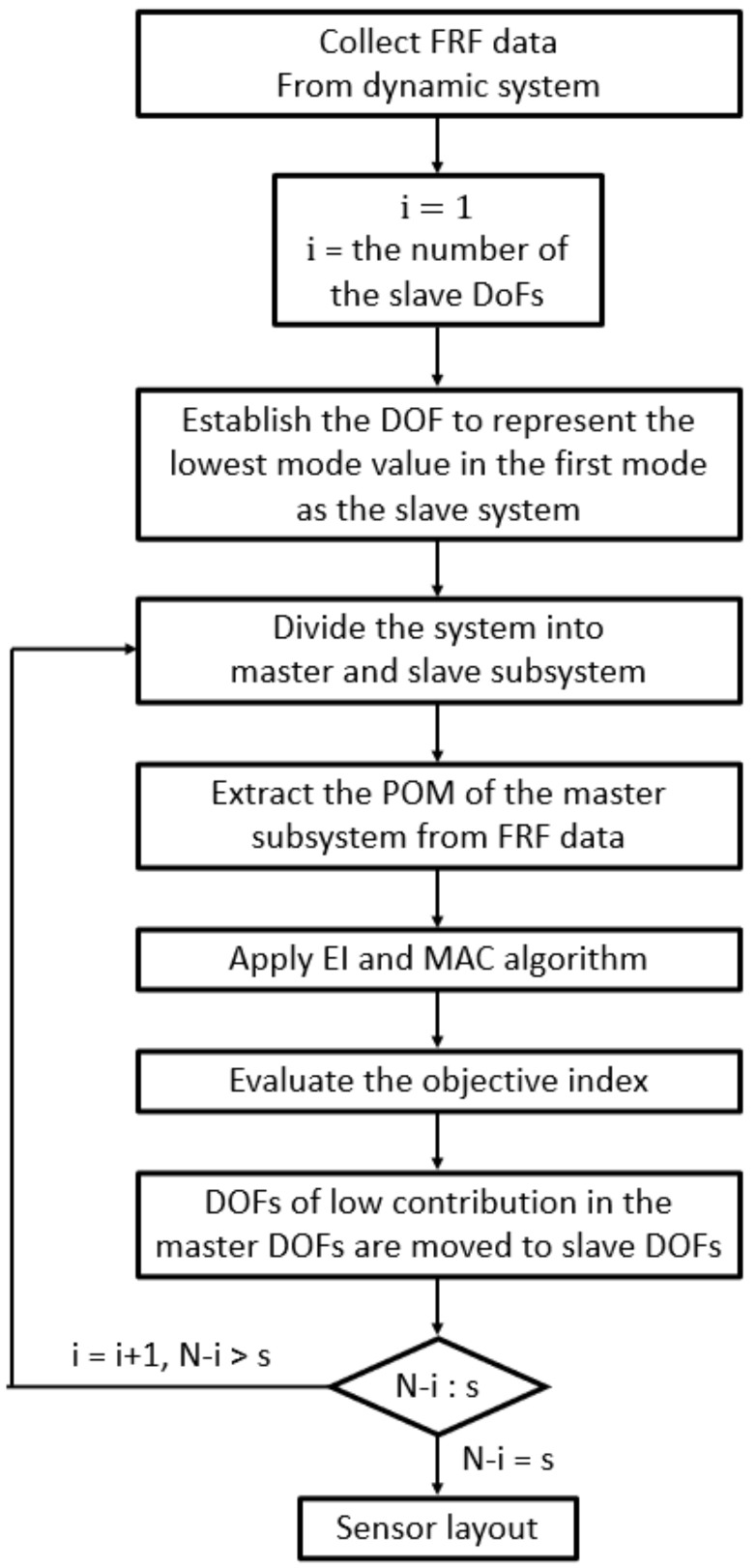
Flow chart of the proposed OSP algorithm.

**Figure 2 sensors-22-03778-f002:**
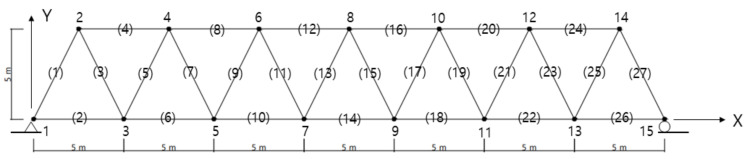
A plane truss (The number in the bracket indicates the member).

**Figure 3 sensors-22-03778-f003:**
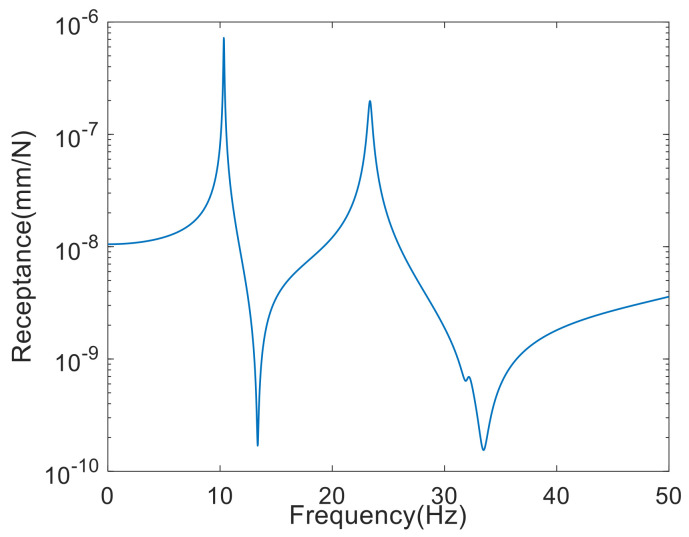
Receptance curve of the truss structure.

**Figure 4 sensors-22-03778-f004:**
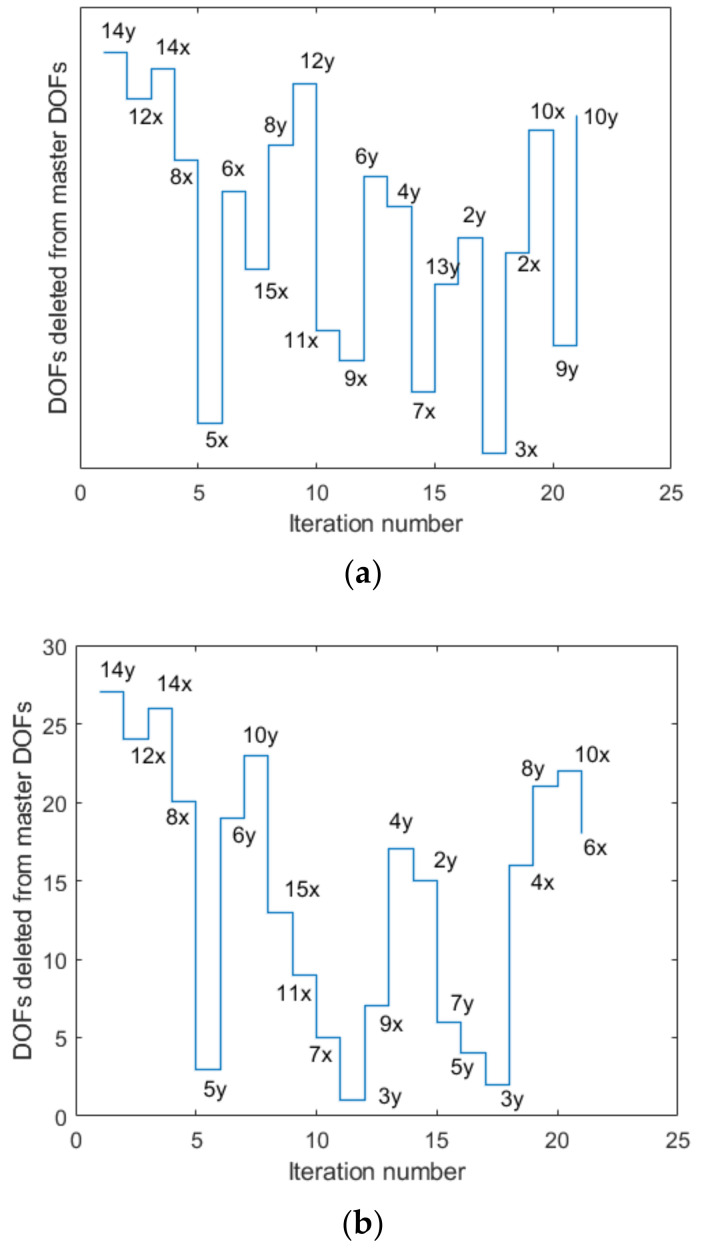
DOFs deleted from master DOFs: (**a**) MAC method; (**b**) EI method.

**Figure 5 sensors-22-03778-f005:**
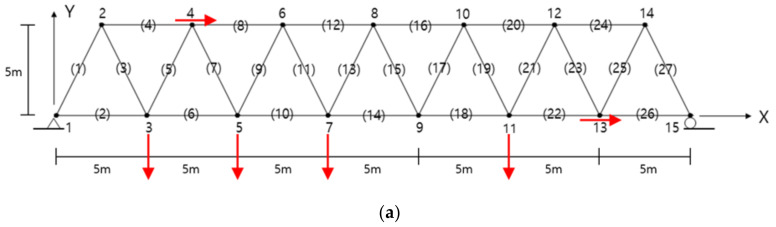
Final sensor layout: (**a**) MAC approach; (**b**) EI approach; (**c**) Sun & Buyukozturk. Numbers in parentheses indicate member and arrows indicate the position and direction of the sensor.

## Data Availability

The data used to support the findings of this study are included within the article.

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
