# Peer review of "An Optimal Sensor Layout Using the Frequency Response Function Data within a Wide Range of Frequencies"

_sensors, 2022, doi:10.3390/s22103778_

Round 1
Reviewer 1 Report
The paper presents a method to optimal sensor layout by using MAC and EI algorithm. These methods can effectively get the best sensor layout. The presented treatment method is novel which is validated by numerical examples. Thus, a minor revision is recommended that includes the following information.
- In the abstract, the meaning of ‘OSP’ should be explained in the text.
- In the keywords, is the existence of sensor necessary?
- In the figure 1, whether the judgment condition can be written as ‘i<=s’.
- In the figure 3, the ordinate is rewritten as m / N, because both figure 2 and density conditions use the unit ‘m’.
- In the figure 4, is there no ordinate in picture in figure (a) and the meaning of ordinates.
Author Response
The paper presents a method to optimal sensor layout by using MAC and EI algorithm. These methods can effectively get the best sensor layout. The presented treatment method is novel which is validated by numerical examples. Thus, a minor revision is recommended that includes the following information.
Thanks for the good point. After careful consideration, the commented points have been revised as follows.

Reviewer 2 Report
(1) In this paper, the frequency response is processed by data dimensionality reduction, and the optimal sensor layout is realized by objective function optimization. The processing method of data dimensionality reduction has been widely used. So, the innovation of the article is lacking.
(2) Is it necessary to normalize the measured data when using POD method to process the data? The frequency with large receptance in the FRF will result in low weight at other frequencies, which will affect the sensitivity of the reduced model to other frequencies.
(3) The interception of the final dimensionality reduction parameter h of POD method will affect the result of selecting the final main degrees of freedom, but the selection basis given in this paper is not clear and convincing.
(4) The analysis of the difference between the final sensor layout using MAC method and EI method is not convincing. Moreover, there are obvious differences between the sensor layout obtained from other documents analyzing the truss structure model. Please analyze the reasons for the differences.
Author Response
Thanks for the good point. After careful consideration on the comments of the reviewer, the raised points have been revised as follows.

Reviewer 3 Report
In this study, an optimal sensor placement framework for structural health monitoring is proposed, based on the proper orthogonal mode extracted from the measured frequency response function data. The article contains numerical tests of a plane truss to compare modal assurance criterion and effective independence algorithm for the sensor locations. However, some clarification and improvements must be attained before publication.
- The reviewer believes that the first time OSP appears, the full name needs to be written in line 9. The same question is in line 252 of the article.
- The introduction section describing the current state of the art of optimal sensor placement should be further described.
- Is there a typo in line 336 of the article describing the location of the sensor mounted at the vertical node?
- The specific meaning of the vertical coordinates in Figure 4 should be written.
- From the numerical example, it can be seen that when 6 sensors are arranged, the MAC and EI methods obtain similar positions of the arranged sensors due to the symmetry of the structure. Is this conclusion universal, and can the same conclusion be obtained when other numbers of sensors are arranged? However, except for the three symmetrical sensors, the other three sensors are not arranged symmetrically. Please explain the difference between the two methods.
- The results and discussions section should present quantitative results and not only the most important qualitative results. Therefore, significant improvements are expected.
7. Please read the instructions on how to describe the references at the end of the article in the authors' guide and change it. Currently, the references at the end of the text are not in line with the journal requirements.
Author Response
In this study, an optimal sensor placement framework for structural health monitoring is proposed, based on the proper orthogonal mode extracted from the measured frequency response function data. The article contains numerical tests of a plane truss to compare modal assurance criterion and effective independence algorithm for the sensor locations. However, some clarification and improvements must be attained before publication.
Thanks for the good point. After careful consideration, the commented points have been revised as follows.

Reviewer 4 Report
Reviewed paper is related to optimisation of sensor placement for vibration based SHM.
In the purpose of sensor placement optimisation the modal assurance criterion and effective independence algorithm were utilised. Proposed algorithms use the proper orthogonal mode extracted from frequency response functions of systems for wide range of frequencies.
Bellow I have listed my comments:
- State of art is very poor, relatively old literature positions are cited.
- I have not understood that experimental technique was also utilised. There are two sentences: “The FRF data from 0.01 Hz to 50 Hz were taken in increments of 0.02 Hz by numerical ” and “A range of measured FRF data was extracted for predicting the POMs by the POD 318 approach. This study utilized 2500 FRF datasets at intervals of 0.02 Hz from 0.01 Hz to 50 319 Hz.”. First is about numerical simulations but second suggest the experiments. There is lack of details of experiment if it was conducted and used.
- Authors use displacements but very often accelerometers are utilised because of larger sensitivity. It is much easier to measure accelerations in the case of small amplitudes of vibrations. Could authors use this methodology for accelerations?
- Comparison of results for two methods in fig. 5 tells nothing. I don’t see any symmetry in this figure. There is vertical direction in fig. 5b (node 12) which is not present in the top part of structure in fig. 5a. Authors need to prove that both sets of optimal sensor locations give the similar set of identified system parameters/signals.
- Results of numerical simulations need to be validated by measurements.
This paper could not be published in present form.
Author Response
Reviewed paper is related to optimisation of sensor placement for vibration based SHM.
In the purpose of sensor placement optimisation the modal assurance criterion and effective independence algorithm were utilised. Proposed algorithms use the proper orthogonal mode extracted from frequency response functions of systems for wide range of frequencies.
Below I have listed my comments:
Thanks for the good point. After careful consideration on the comments of the reviewer, the raised points have been revised as follows.

Round 2
Reviewer 2 Report
For question 3, it is necessary to pay attention to the selection of the number h of POMs in the application of this method. Different h may lead to differences in the final results of OSP.
For question 4, are the OSP results based on FRF and POD obtained in this paper comparable with the results obtained in the original article of the truss structure model(Sun and Buyukozturk)? Only comparing the two groups of results obtained in this paper can not completely prove the effectiveness of the proposed method.
Author Response
Thank you again for your invaluable comment.
For question 3, it is necessary to pay attention to the selection of the number h of POMs in the application of this method. Different h may lead to differences in the final results of OSP.
-à Thanks for the good point. It is important to choose the first slave DOF. In this study, the slave DOF representing the smallest value of the mode shape was selected as the slave DOF, and the DOF with a low contribution was moved from the master DOF to the slave DOF at each iteration after that, and the final sensor position was estimated. Initially, if the slave DOF is selected as the slave DOF at the position where the value of the mode shape is small, the iteration process is slightly different, but the same OSP can be obtained.
In the previous study1), in order to set the positions of the two and four final sensors in the example of estimating the final sensor positions, the first two and four mode shape matrices were used for calculations, respectively. As a result, we could see that the positions representing the two sensors are contained within the four sensor positions. From this, it can be inferred that the same result would be obtained if the operation as described above was repeated and executed.
- Lee, Eun-Taik, Eun, Hee-Chang, 2021, Optimal sensor placements using modified Fisher information matrix and effective information algorithm, International Journal of Distributed Sensor Networks, 17(6), doi:10.1177/15501477211023022.
For question 4, are the OSP results based on FRF and POD obtained in this paper comparable with the results obtained in the original article of the truss structure model(Sun and Buyukozturk)? Only comparing the two groups of results obtained in this paper can not completely prove the effectiveness of the proposed method.
-à Figure 5(c) was added. The final OSPs were compared with the numerical results by Sun & Buyukozturk. They took 8 DOFs as the OSPs for the numerical example. Compared with the results of this study, two OSP algorithms in this study exhibited the same sensor positions in 4 DOFs. This difference is interpreted to be due to the difference in the algorithm including the location of the objective function.
Figure 5(c) shows the OSPs obtained from the example provided by Sun & Buyukozturk. That example estimated the positions of 8 sensors. Compared with the results of this study, two OSP algorithms in this study exhibited the same sensor positions in 4 DOFs. This difference in the OSPs is interpreted to be due to the difference in the algorithm including the location of the objective function.
Reviewer 3 Report
The author revised the manuscript according to the review comments.
Author Response
The author revised the manuscript according to the review comments.
-à Thank you for your comment.
Reviewer 4 Report
Dear Authors thank you very much for explanation. Paper was slightly improved, but as far as it is based on exclusively numerical data, with only one example of structure it cannot be published. Authors should validate numerical results using experimental part. In case it is not possible more cases of structures should be investigated.
Author Response
Dear Authors thank you very much for explanation. Paper was slightly improved, but as far as it is based on exclusively numerical data, with only one example of structure it cannot be published. Authors should validate numerical results using experimental part. In case it is not possible more cases of structures should be investigated.
-à
Thanks for the very good point.
As I have answered before, it will take too much time to do the experiment, and it is difficult to do another numerical experiment. However, using the current algorithm, we will verify more theoretical approaches and validity in the future and prove it through various numerical experiments.
Instead of conducting experiments or numerical analysis, the validity of the results of this study was discussed by comparing them with the OSP obtained by Sun & Buyukozturk. It was inserted in the manuscript. Compared with the results of this study, two OSP algorithms in this study exhibited the same sensor positions in 4 DOFs. This difference in the OSPs is interpreted to be due to the difference in the algorithm including the location of the objective function.